# Systemic Inflammation at the Crossroad of Major Depressive Disorder and Comorbidities: A Narrative Review

**DOI:** 10.3390/ijms26199382

**Published:** 2025-09-25

**Authors:** Erika Vitali, Nadia Cattane, Ilari D’Aprile, Giulia Petrillo, Annamaria Cattaneo

**Affiliations:** 1Biological Psychiatry Unit, IRCCS Istituto Centro San Giovanni di Dio Fatebenefratelli, 25125 Brescia, Italy; evitali@fatebenefratelli.eu (E.V.); ncattane@fatebenefratelli.eu (N.C.); or idaprile@fatebenefratelli.eu (I.D.); or gpetrillo@fatebenefratelli.eu (G.P.); 2Department of Pharmacological and Biomolecular Sciences, University of Milan, 20133 Milan, Italy

**Keywords:** major depressive disorder, inflammation, comorbidities, cardiometabolic disorders, autoimmune disorders, chronic pain disorders

## Abstract

Major Depressive Disorder (MDD) represents a global challenge due to its high prevalence worldwide. Inflammation is the most extensively studied and plausible biological pathway involved in the onset of MDD. Individuals with MDD often exhibit low-grade inflammation, characterized by immune system dysregulation and activation of pro-inflammatory pathways. Elevated inflammation is also associated with a reduced response to antidepressant therapies, suggesting that targeting inflammation could represent a promising therapeutic approach for MDD. MDD frequently co-occurs with other pathological conditions, including cardiometabolic, autoimmune, and chronic pain disorders. These comorbidities further complicate MDD treatment and contribute to reduced antidepressant efficacy. Like MDD, these disorders are characterized by a strong inflammatory component, and several cytokines and pro-inflammatory mechanisms altered in MDD are also found in these comorbid conditions. This narrative review explores inflammation as a shared biological mechanism in MDD and its most frequent comorbidities, to provide a comprehensive understanding of the interplay between inflammation and these comorbid conditions. Persistent low-grade inflammation may help explain the high rate of bidirectional co-occurrence between MDD and its comorbidities. Moreover, it may represent a target for better understanding the molecular mechanisms driving this co-occurrence, potentially contributing to the development of tailored treatment and improving antidepressants response rates.

## 1. Introduction: Role of Inflammation in the Pathophysiology of Major Depressive Disorder

Major Depressive Disorder (MDD) is a severe psychiatric disease characterized by several clinical symptoms, including depressed mood, loss of interest or pleasure in activities, sleep disturbances, and fatigue [1]. Affecting approximately 300 million people worldwide, MDD is among the leading causes of global disease burden and suicide [2]. Interestingly, females have a two-fold higher risk of developing MDD than males, probably due to sex-based biological differences, including sex hormones, brain structure, the immune system, inflammation, and metabolism [3].

Several hypotheses have been proposed to explain the pathophysiology of MDD; however, none of them alone can fully account for the complex biological basis of this disorder [4]. The inflammatory hypothesis is one of the most studied mechanisms underpinning MDD and it is strongly supported by evidence that confirms its involvement in both MDD onset and antidepressant (AD) treatment response, and it has also been implicated in several common comorbidities associated with MDD. As reported in the meta-analysis by Osimo and colleagues, low-grade inflammation, defined as a C-reactive protein (CRP) level between 3 and 10 mg/L, occurs in approximately 30% of MDD patients [5,6], while about 60% of patients exhibit mildly elevated CRP levels (> 1mg/L). The role of inflammation in MDD pathophysiology is further supported by changes in the expression levels of several inflammation-related genes (Table 1). Indeed, depressed patients show an up-regulation of several pro-inflammatory genes in peripheral blood, including Interleukin 1β (*IL*-*1*β), Interleukin 1 Receptor Type1 (*IL*-*1R1*), TNF Receptor Superfamily Member 1A (*TNFR1*), *TNFR2*, *IL*-*6*, C-C Motif Chemokine Ligand 2 (*CCL2*), and Tumor Necrosis Factor (*TNF*-α), whereas anti-inflammatory genes, for instance *IL*-*4*, and genes involved in transduction of anti-inflammatory cascade, such as glucocorticoid receptor (*GR*), are downregulated [7,8,9,10]. Similarly, the expression of cytokines signaling suppressors, such as Suppressor Of Cytokines Signaling (*SOCS*) 1, *SOCS2*, and *SOCS3*, is reduced in MDD [11]. Modulation of mRNA levels of inflammation-related genes may also reflect disease severity: for instance, *CRP* and *TNF*-α levels positively correlate with Montgomery-Åsberg Depression Rating Scale (MADRS) score, suggesting that higher inflammatory levels are associated with more severe depressive symptoms [11,12]. At the protein level, the concentration of several cytokines and inflammation-related molecules is altered in the blood of MDD patients (see Table 1 for details) [10,13,14,15,16,17]. Furthermore, levels of IL-1β, IL-10, TNF-α, CCL4, and CCL8 proteins are positively associated with depressive symptoms assessed by the Hamilton Depression Rating Scale (HDRS) [13,16,17,18].

Another inflammation-related mechanism implicated in MDD is the kynurenine pathway, which metabolizes tryptophan into several intermediate metabolites, ultimately leading to the production of nicotinamide adenine dinucleotide phosphorylated NAD(P)+ [19]. A dysregulation of this pathway, often triggered by elevated levels of pro-inflammatory molecules, leads to the accumulation of neurotoxic metabolites, such as the quinolinic acid, thereby promoting inflammation [19].

**Table 1 ijms-26-09382-t001:** Inflammation-related molecules involved in MDD and comorbid conditions.

Disease		Inflammation-Related Proteins	References	Inflammation-Related Genes	References
**Major Depressive Disorder**		Upregulated: CCL3; CCL4; CCL8; CCL11; CRP; IL-1β; IL-2; IL-6; IL-8; IL-10; IL-12; IL-13; IL-15; TNF-α	[5,8,9,13,16,17,18,20]	Upregulated: *CCL2*; *FKBP5*; *GR*; *IL*-*1β*; *IL*-*1R1*; *IL*-*6*; *TNF*-*α*; *TNFR1*; *TNFR2*	[7,9,10,11]
		Downregulated: CCL17; IL-5; IL-16; TNF-β		Downregulated: *GR*; *IL4*; *SOCS1*; *SOCS2*; *SOCS3*	
**Cardiometabolic Diseases**	Heart Failure	Upregulated: hs-CRP; IL-1β; IL-6; IL-8; TNF-α	[21,22,23,24,25]	Upregulated: *TNF*-*α*	[25]
	Ischemic Heart Disease	Upregulated: IL-6	[26]	Upregulated: *IL*-*1*	[27]
	Coronary Arthery/Heart Disease	-	-	Upregulated: *TNF*-*α*	[28]
	Myocardial Infartion	Upregulated: CRP; IL-6; IL-8; TNF-α	[29,30]	Upregulated: *TNF*-*α*	[31]
	Hypertension	Upregulated: CRP; hs-CRP; IL-6; TNF-α	[32]	-	-
	Ischemic Stroke	Upregulated: CRP; IL-6; IL-8; TNF-α	[33,34,35,36]	-	-
	Diabetes	Upregulated: hs-CRP; IL-6; TNF-α	[37,38,39]	Upregulated: *CCL2*; *CRP*; *IL*-*1β*; *IL*-*6*; *TNF*-*α*	[40]
**Autoimmune Diseases**	Rheumatoid Arthritis	Upregulated: IL-1β; IL-6; TNF-α	[41,42,43,44]	Upregulated: *TNF*-*α*	[44]
	Systemic Lupus Erythematosus	Upregulated: IFN-γ; IL-12; IL-17; IL-23; TNF-α	[45]	-	-
	Psoriasis Vulgaris	Upregulated: IL-6; IL-17; TNF-α	[41]	-	-
	Multiple Sclerosis	Upregulated: CCL4; CCL22; CXCL10; IL-6; IL-12 Downregulated: IL-10	[46,47]	Upregulated: *IFN*-*γ*; *TNF*-*α*	[48]
				Downregulated: *IL*-*10*	
**Chronic Pain-Related Diseases**	Low Back Pain	Upregulated: CRP; IL-1β; IL-6; TNF-α	[49,50]	-	-
	Fibromyalgia	Upregulated: hs-CRP; IL-6; IL-8; TNF-α	[51,52]	-	-
	Migraine	Upregulated: IL-1β; IL-6; IL-8 TNF-α	[53]	-	-
	Chronic Fatigue/Myalgic Encephalomyelitis	Upregulated: CCL24; CXCL-10; hs-CRP; IFN-γ; IL-7; IL-8; TNF-α	[54,55,56,57]	-	-
	Endometriosis	Upregulated: IL-1;IL-2; IL-8; IL-33; TNF-α	[58]	-	-

Besides peripheral inflammation, MDD is also characterized by neuroinflammation, involving the activation of microglia, which shifts from a resting state to an activated state in response to infections or insults, thus releasing pro-inflammatory cytokines [59]. Neuroinflammation levels are associated with MDD severity: indeed, microglial activation in several brain regions is positively correlated with HDRS scores [60,61]. 

Microglial activation is also closely related to another mechanism implicated in MDD, namely the hypothalamic–pituitary–adrenal (HPA) axis [20]. Under physiological conditions, the HPA axis is activated in response to stress, which induces the release of corticotropin-releasing hormone (CRH) from the hypothalamus and, consequently, the secretion of adrenocorticotropic hormone (ACTH) from anterior pituitary gland. ACTH, in turn, stimulates the release of glucocorticoids (GCs), in particular cortisol, from the adrenal cortex. Cortisol acts through a negative feedback loop, inhibiting further secretion of CRH and ACTH by binding to GRs in the hypothalamus and pituitary gland [62]. Under chronic stress, this mechanism is disrupted and the continuous production of cortisol desensitizes GRs, leading to GC resistance and aberrant HPA axis activation [62]. In the presence of neuroinflammation, microglial activation can further stimulate HPA axis activity. Indeed, pro-inflammatory cytokines released by microglia, including IL1-β, IL-6, IFN-α and TNF-α, triggers CRH secretion from the hypothalamus, resulting in HPA axis hyperactivation [20]. In turn, the release of GCs may contribute to neuroinflammation through the binding of cortisol to mineralocorticoid receptors and GRs expressed on the microglia. In particular, excessive levels of GCs promote the pro-inflammatory activation of GR-rich microglial cells, thus sustaining the neuroinflammatory status [20]. 

Another mechanism by which stress may induce neuroinflammation involves the inflammasome complex. When stressed by pathogen- or damage- associated molecular patterns, brain cells can trigger the interaction of Nod-like receptor protein 3 (NLRP3) with the apoptosis-associated speck-like protein (ASC), which recruits pro-caspase-1, thereby assembling the inflammasome complex and activating caspase-1. Caspase-1, in turn, cleaves and activates IL-1β and IL-18, thus promoting the neuroinflammatory response [63]. Interestingly, NLRP3 inflammasome expression is elevated in the prefrontal cortex and in peripheral blood cells of MDD patients who died by suicide compared to healthy individuals [64,65]. However, peripheral NLRP3 levels may not fully reflect the neuroinflammation status, as NLRP3 is also expressed by peripheral cells [66]. A promising peripheral biomarker of neuroinflammation may be represented by IL-34 found in neuron-derived exosomes, whose levels are significantly higher in the plasma of MDD patients compared to healthy controls [67]. Similarly, TNFR1 in neuron-derived exosome is positively correlated with the Beck Depression Inventory II score and with several clinical symptoms, including affective, motivational, cognitive, behavioral, and vegetative dimensions [67]. 

Inflammation may also help to explain the higher prevalence of MDD in females. Indeed, females tend to have higher levels of inflammation than males and experience more frequently mood and behavioral changes induced by inflammation [68]. Sex differences have also been observed in inflammation-related biomarkers: females with MDD showed elevated levels of CRP, IL-6, IL-1β, which are not increased in males [69], while males with MDD showed increased levels of IL-17 [70]. Finally, only females have been shown to experience increased depressed mood following an inflammatory stimulus, characterized by correlations between IL-6 and TNF-α levels and social disconnection [71], as well as between CRP levels and depression severity [12,70]. Moreover, the neutrophil-to-lymphocyte ratio, an inflammatory marker found to be elevated in females but not in males [72], has been shown to correlate negatively with HDRS scores in females and positively in males [73].

## 2. Antidepressant Treatments: How the Inflammation May Modulate and Predict Their Efficacy

Systemic inflammation may represent a valid biomarker predicting ADs treatment outcome. Indeed, elevated levels of certain pro-inflammatory markers have been associated with reduced responsiveness to AD treatments in MDD, whereas resistance to multiple treatments is predictable by high CRP levels and increased expression of inflammation-related genes [7,74]. Furthermore, a reduction in *SOCS3* mRNA levels has been observed among depressed patients who achieved remission following AD treatment [11], while high Macrophage Migration Inhibitory Factor (*MIF*) levels have been associated with less symptom improvement [75]. Concentration of circulating cytokines has been proposed as potential diagnostic and predictive biomarker for MDD and AD response. For instance, serum levels of IL-1α, IL-5, Intercellular Adhesion Molecule 1, and IL-1β have been shown to predict AD treatment response and MDD diagnosis with high accuracy [17,75,76], thus supporting the clinical utility of peripheral immune-inflammatory biomarkers as predictors of treatment efficacy. Furthermore, AD treatment can modulate inflammatory status. Indeed, patients who successfully respond to AD therapy exhibit reduced protein level of MIF, as well as decreased expression of FKBP prolyl isomerase 5 (*FKBP5*), *IL*-*6*, and *IL*-*1*β alongside increased *GR* mRNA level [7,75]. Notably, the specific inflammatory molecules modulated by ADs differ depending on the drug class. For example, fluoxetine, a selective serotonin reuptake inhibitor (SSRI), has been observed to decrease IL-6 serum levels, while amitriptyline, a tricyclic AD, reduces *NLRP3* and *caspase*-*1* expression, as well as IL-1β and IL-18 serum levels [65,77]. 

According to a growing line of evidence supporting the role of inflammation in MDD and the anti-inflammatory effects of ADs, several clinical trials are currently investigating the potential therapeutic role of anti-inflammatory agents, both pharmacological and non-pharmacological, in MDD. For instance, recent studies have explored the use of nonsteroidal anti-inflammatory drugs, such as celecoxib, in patients with inflammatory or metabolic dysregulations, aiming to determine whether adjunctive treatment leads to better outcomes than AD therapy alone [78] and whether it can reduce neuroinflammation when used as monotherapy (ClinicalTrial.gov ID: NCT04814355). Similarly, the antibiotic minocycline, known for its anti-inflammatory properties, is under investigation in MDD patients with comorbid obesity (ClinicalTrial.gov ID: NCT06537921). Interestingly, the response to minocycline can be predicted by baseline CRP levels in a sex-dependent manner: in particular, CRP levels ≥ 3 mg/L predict treatment response only in females, who are also the only ones to show a reduction in CRP levels after treatment [79]. In addition, the adjunction of N-acetylcysteine to AD therapy is being evaluated in treatment-resistant depression (TRD) patients with elevated peripheral inflammation for its potential effects on brain function, reduction of inflammatory biomarkers, and an improvement in white matter integrity (ClinicalTrial.gov ID: NCT02972398). Inhibition of inflammatory cytokines has also been evaluated as a strategy to improve depressive symptoms. For instance, the adjunctive use of infliximab, a monoclonal antibody targeting TNF-α, improved HAMD-17 scores in patients with hs-CRP > 5 mg/L, and decreased inflammation levels [80]. Similarly, adalimumab, a monoclonal antibody that binds to TNF-α, when combined with sertraline, induced a reduction of HAM-D score in MDD patients [81]. Also, etanercept and tocilizumab, already used in autoimmune diseases, are able to improve depressive symptoms in patient with psoriasis and rheumatoid arthritis, respectively [82,83,84]. In addition to pharmacological anti-inflammatory agents, natural compounds with strong anti-inflammatory properties, such as curcumin, have also been investigated. Curcumin, a compound derived from the Curcuma longa plant, has interesting anti-inflammatory properties and can modulate neurotransmitter concentration, inflammatory pathways, neuroplasticity, HPA axis impairment, and other mechanisms involved in MDD [85]. Notably, MDD patients treated with curcumin showed a decrease in both HDRS-17 and MADRS score compared with the placebo group, along with a reduction of pro-inflammatory cytokines [86].

Encouraging evidence for the efficacy of anti-inflammatory agents in the treatment of MDD has emerged from recent clinical trials. These studies have demonstrated improved outcomes in patients receiving combination therapy compared to AD monotherapy [87], as well as reduced depressive symptoms among TRD patients with high baseline inflammation (ClinicalTrial.gov ID: NCT00463580) [80]. Moreover, an ongoing trial is assessing the effects of anti-inflammatory treatment alone in MDD patients with low-grade inflammation (ClinicalTrial.gov ID: NCT06136546), without the use of concomitant AD therapy. Indeed, as comprehensively reviewed by Du and colleagues, the administration of anti-inflammatory drugs may represent a valid therapeutic strategy for MDD, even when not combined with conventional ADs [88]. Finally, several clinical trials have also evaluated the utility of omega-3 fatty acids for the treatment of MDD, due to their anti-inflammatory properties [89]. These include studies conducted on different cohorts of patients, such as adolescents, adults, and pregnant and/or breastfeeding females (ClinicalTrial.gov ID: NCT00511810; NCT00962598; NCT00289484; NCT00618865). Other trials have investigated the use of different omega-3 fatty acids and the personalization of the treatment based on clinical subtypes (ClinicalTrials.gov ID: NCT00361374; NCT03732378; NCT03871088). For instance, a clinical trial evaluated patients according to their inflammation levels (ClinicalTrial.gov ID: NCT03143075). This approach revealed that the combination of omega-3 fatty acids with ADs leads to a significant greater improvement in depressive symptoms compared to ADs alone, particularly among patients with higher levels of inflammation [90,91]. Moreover, the meta-analysis conducted by Liao and colleagues provided encouraging results regarding the use of omega-3 fatty acids in the treatment of MDD [92].

## 3. Inflammation as a Triggering Mechanism of Comorbidities in MDD

Clinical management of MDD is further complicated by the frequent presence of comorbid conditions, which increase treatment complexity and are often associated with poorer therapeutic outcomes [70]. These comorbidities also contribute to the economic burden of MDD. Indeed, approximately 40% of MDD total costs are attributable to direct and indirect costs related to co-occurrent diseases [93]. 

MDD is associated with an increased incidence of several comorbid conditions, including both physical (e.g., cardiovascular, metabolic, respiratory, oncological, neurological, gastroenterological, autoimmune, musculoskeletal, and chronic pain-related disorders) and psychiatric disorders (e.g., anxiety, eating and sleeping disorders, dysthymia, substance abuse, schizophrenia, and antisocial personality) [94,95,96,97], which may interfere with AD treatment response [70]. Interestingly, the presence of MDD in adulthood may predict the onset of these disorders later in life. For instance, during aging, MDD patients have an increased risk of developing neurological disorders such as Parkinson’s disease and Alzheimer’s disease [98,99], as well as cardiovascular conditions [100]. Additionally, late-life onset of MDD has been associated with a higher risk of several aforementioned co-occurrent conditions [101,102,103], which may contribute to elevated mortality in patients with more severe depressive symptoms [104].

These comorbidities may be explained by several shared biological and environmental risk factors, among which inflammation plays a pivotal role [105]. This narrative review explores the most frequent conditions co-occurring with MDD, namely cardiometabolic, autoimmune, and chronic pain-related disorders, and focuses on the role of inflammation as a shared biological mechanism, aiming to provide a comprehensive understanding of the interplay among inflammation, MDD, and its comorbidities (Figure 1). In particular, we will discuss inflammation-related molecules involved in MDD comorbidities to elucidate the presence of both shared and condition-specific inflammation-related biomarkers.

### 3.1. Alterations in Cytokine Patterns Are Associated with the Development of Several Cardiometabolic Disorders and Their Co-Occurrence with MDD

Cardiometabolic diseases (CMDs) encompass several illnesses having a cardiovascular (CVDs) or a metabolic involvement, such as heart failure (HF), ischemic heart disease (IHD), coronary artery/heart disease (CAD/CHD), myocardial infarction (MI), type 1 (T1D), and type 2 (T2D) diabetes. 

#### 3.1.1. Cardiovascular Disorders

CVDs are the leading cause of mortality worldwide and affect about 500 million people globally [106]. The comorbidity between MDD and several CVDs is well-established and multifactorial, and it can be explained by the involvement of metabolic, inflammatory, psychosocial, and lifestyle-related risk factors [107]. Furthermore, MDD is not only more prevalent among individuals who have experienced major cardiac events compared to the general population [108], but it also predicts worse clinical outcomes, such as increased mortality and higher rates of recurrence [109]. For example, approximately 29% of post-MI patients develop clinically significant depressive symptoms [110], which are associated with a two-fold increased risk of mortality within one year [111]. Similarly, post-stroke depression affects a comparable percentage of patients, with a cumulative incidence reaching up to 42% over five years and it is linked to increased disability and mortality [112,113]. MDD also increases the likelihood of developing hypertension [114] and, vice versa, hypertensive patients have increased risk of developing MDD [115]. Furthermore, MDD affects up to 60% of HF patients and represents an independent predictor of mortality and hospital readmissions [106,116]. Interestingly, these associations appear to be stronger in females than in males potentially due to hormonal influences [117], greater psychosocial stress exposure, and gender-specific symptom presentations. Although contributing factors include behavioral risk (smoking, inactivity) and physiological changes (endothelial dysfunction, lipid dysregulation) [118], chronic low-grade inflammation, which plays a central role in CVDs, has emerged as a key pathophysiological link underlying this comorbidity [117]. Indeed, elevated levels of peripheral inflammatory mediators have been observed across various CVDs [119,120,121,122,123]. Particularly, in patients with HF high plasma concentration of high sensitivity CRP (hs-CRP) protein have been reported [21], as well as elevated levels of different interleukins, including IL-1, IL-6, and IL-8 [22,23,24], and increased protein and mRNA levels of *TNF*-*α* in cardiac post-mortem tissue [25]. Similarly, IHD patients show high IL-6 levels [26] and increased mRNA expression of *IL*-*1* [27]. Genetically-predicted TNF-α levels has been associated with CHD [28], whereas elevated peripheral levels of IL-8, TNF-α, CRP, and IL-6 have been reported in MI [29,30,31]. Moreover, CRP, hs-CRP, and IL-6 have been associated with an increased risk of developing hypertension, whereas TNF-α has been linked to elevated blood pressure in hypertensive patients [32]. Finally, both IL-6 and IL-8 have been associated with an increased risk of ischemic stroke, with IL-8 levels showing a positive correlation with the severity of post-stroke disability [33,34]. 

These overlapping inflammatory profiles between MDD and CVDs suggest that systemic inflammation may act as a shared biological substrate underlying their co-occurrence. From a cardiovascular perspective, chronic elevation of cytokines, such as IL-6, promotes a pro-atherogenic environment [124] and stimulates the hepatic production of CRP [125], which is a strong independent predictor of adverse cardiovascular events [35]. Another cytokine, TNF-α, on the other hand, directly impairs endothelial nitric oxide production, promoting vasoconstriction, oxidative stress, and thrombosis, hallmarks of endothelial dysfunction and cardiovascular instability [36]. Beyond their cardiovascular effects, these same cytokines can reach the brain through several routes and impact brain function influencing neurobiological pathways involved in mood regulation [126].

Taken together, the shared inflammatory profile observed in both MDD and CVDs partially explains why individuals with chronic cardiovascular conditions are more susceptible to developing depressive symptoms, and vice versa [117]. Moreover, some evidence suggests that ADs may act as an independent risk factor for CVDs and may potentially exacerbate cardiovascular symptoms in at risk patients. Therefore, it is crucial to carefully consider both the potential beneficial and detrimental effects of ADs treatment in MDD patients with cardiovascular comorbidities [127]. Furthermore, inflammation-related biomarkers involved in CVDs are differently modulated in males and females, and this should be taken into account when evaluating the impact of inflammation on ADs response in patients with CVDs. For instance, CRP levels have been found to predict myocardial infarction, stroke, and cardiovascular death only in females [128,129,130,131].

Finally, this evidence suggests that therapeutic strategies aimed at modulating inflammation, such as anti-inflammatory drugs, cytokine inhibitors, or lifestyle interventions targeting immune function, could hold promise for improving both cardiovascular and mental health outcomes in patients affected by these comorbid conditions.

#### 3.1.2. Diabetes

Another frequent co-occurrent disorder in MDD patients is diabetes. Together with MDD, diabetes is among the most prevalent and impactful non-communicable diseases worldwide, and their frequent co-occurrence has become an increasing focus of clinical and research studies [132]. Diabetes, encompassing both T1D and T2D, affects more than 537 million adults worldwide, and it shows a rising global prevalence [133,134]. Individuals with diabetes are nearly twice as likely to experience depressive symptoms or develop MDD compared to the general population, and, conversely, those with MDD face an elevated risk of developing diabetes [135,136]. However, this comorbidity is not merely additive in its effects. Patients affected by both conditions tend to exhibit more severe clinical outcomes, including poorer glycemic control, higher rates of diabetes-related complications such as CVDs and nephropathy, diminished quality of life, and increased mortality [137,138]. Furthermore, comorbid MDD and T2D are associated with significantly higher healthcare utilization and costs [139,140], highlighting the urgent need for a deeper understanding of the biological mechanisms that underlie this bidirectional relationship. In the context of diabetes, particularly when accompanied by obesity or metabolic syndrome, adipose tissue acts as an active endocrine organ, releasing pro-inflammatory cytokines that exacerbate systemic inflammation, above all CCL2, IL-1β, and IL-6 [40]. This chronic low-grade inflammatory state may extend to the central nervous system and contribute to neurobiological changes implicated in depressive symptomatology [141], representing a potential shared pathophysiological mechanism linking MDD and diabetes [137,142]. Inflammatory processes are central to the development and progression of diabetes, particularly through their role in insulin resistance [143,144], β-cell dysfunction [145] and endothelial damage [146]. Elevated levels of pro-inflammatory cytokines such as CRP, IL-6, and TNF-α have been observed in both individuals with diabetes and those with depressive symptoms [37,38]. Recent studies have further elucidated the role of these inflammatory markers in the context of comorbidity. For instance, a systematic review and meta-analysis by Nguyen and colleagues reported that individuals with both T2D and depressive symptoms exhibit significantly higher circulating levels of IL-6 and CRP compared to those with T2D alone, suggesting that inflammation may mediate the severity or presence of depressive symptoms in this subgroup [39]. This finding underscores the importance of pro-inflammatory cytokines not only as biomarkers but also as potential mechanistic mediators in the bidirectional relationship between diabetes and MDD. Importantly, cytokines such as IL-6 and TNF-α may activate the HPA axis, leading to sustained hypercortisolemia, a neuroendocrine profile frequently observed in both MDD [147] and poorly controlled diabetes [148]. These neuroimmune interactions may contribute to behavioral symptoms like anhedonia, fatigue, and cognitive dysfunction, commonly reported in patients affected by both conditions [135,149]. Moreover, elevated cytokine levels have been associated with reduced efficacy of AD treatments, particularly SSRIs, in patients with coexisting metabolic disturbances, reinforcing the notion that inflammation plays a role not only in the disease onset but also in the treatment response [74,150,151]. These findings underscore the importance of early identification and comprehensive management of both diabetes and MDD, particularly in individuals with a high inflammatory burden, and suggest that targeting inflammation may hold promise for improving outcomes in this vulnerable population. Similarly to CVDs, also diabetes showed a sex-specific modulation of inflammation-related biomarkers. For instance, baseline CRP levels correlate more strongly with several metabolic indexes, such as dyslipidemia, hypertension, and diabetes in females than males [152].

### 3.2. Immune System Dysregulation and Elevated Levels of Pro-Inflammatory Cytokines Are Shared Biomarkers Between Autoimmune Diseases and MDD

A growing body of evidence suggests a bidirectional association between MDD and autoimmune diseases, such as rheumatoid arthritis (RA), systemic lupus erythematosus (SLE), psoriasis vulgaris (PV), and multiple sclerosis (MS) [153]. Autoimmune diseases are characterized by a reduction or loss of immunological tolerance to self-antigens, that leads to chronic inflammation [154]. Patients suffering from autoimmune diseases are at higher risk of developing MDD, while MDD patients show increased levels and reactivity of autoantibodies [155,156]. Interestingly, higher degrees of autoimmune dysregulation have been observed more in TRD patients than in responders [157]. This bidirectional relationship is probably the result of increased inflammatory activity, characterized by higher levels of pro-inflammatory cytokines, chemokines, and other inflammatory agents [41]. Interestingly, most autoimmune diseases are more prevalent in females than in males, with approximately 78% of all autoimmune diseases occurring in females [158]. Moreover, females with an autoimmune disorder have a two-fold higher risk of developing MDD compared to general population [159]. This may be attributable to the role of sex hormones: indeed, estrogens are potential stimulators of autoimmunity, whereas androgens have an anti-inflammatory and neuroprotective effect [160]. 

#### 3.2.1. Rheumatoid Arthritis

Both MDD and RA show a very high comorbidity rate, with inflammation being the common denominator observed in both diseases [161]. Indeed, patients with MDD are at increased risk of developing RA [162] and, vice versa, patients with RA are exposed to a higher risk of depressive symptoms than the general population [163]. RA affects approximately 1% of the world population and it is among the most common chronic autoimmune inflammatory diseases [164]. It is characterized by stiffness, active arthritis, joint pain, joint erosion, and cartilage damage leading to articular destruction and functional decline [42]. Although the full aetiology and pathogenesis of RA remain unknown, a dysregulation of both the innate and adaptive immune systems have been proposed as a major driver of the disease [165]. Interestingly, various cytokines are implicated in both RA and MDD. For example, IL-1β levels are increased in the synovium of patients with RA compared to control subjects [43]. Similarly, IL-6, which stimulates vascular endothelial growth factor, promoting angiogenesis and the release of autoantibodies through the stimulation of T and B cells in the synovium of RA patients, is increased in the serum of depressed patients [42]. Another important cytokine whose increased levels have been found both in patients with RA and MDD is TNF-α, which enhances the production of both IL-1β and IL-6 [41]. Moreover, higher TNF-α levels have been observed in the synovium of patients with RA [44]. All these studies confirm previous hypotheses about common molecular mechanisms of both RA and MDD and suggest that several RA-related inflammatory markers, above all IL-1β, IL-6 and TNF-α, identified in patients with MDD, can be used as new biomarker candidates or potential therapeutic targets for both the disorders [164]. 

#### 3.2.2. Systemic Lupus Erythematosus

A bidirectional link similar to that observed in RA can be found between MDD and SLE, a chronic and progressive disorder affecting multiple organs with a wide range of clinical manifestations. In the course of this debilitating disease, a dysfunction of both innate and adaptive immune systems has been suggested [166]. The main mechanism of SLE is represented by an excessive immune response of immune cells to autoantigens, which leads to systemic inflammation and inflammation-induced organ damage [166]. Deregulated levels of T helper type 1 (Th1), type 2 (Th2), and type 17 (Th17) cytokines have been shown in patients with SLE compared to control subjects and associated with disease activity and severity [167]. Specifically, during advanced stages of the disease, an ongoing Th1 response is observed, characterized by elevated levels of IL-12, IL-17, IL-23, TNF-α, as well as IFN-γ [45]. Interestingly, the prevalence of MDD in patients suffering from SLE is two times higher compared to the general population, with their quality of life significantly affected [168]. Most of the pro-inflammatory cytokines deregulated in patients with SLE have also been found in depressed patients, suggesting inflammation as a common biological mechanism underlying both of the disorders.

#### 3.2.3. Psoriasis Vulgaris

PV is a chronic, multisystemic, immune-mediated and recurrent dermatosis with unknown etiology, which affects 2–3% of the worldwide population [169]. Based on screening studies, MDD may affect up to 55% of patients with psoriasis, while the incidence of mood disorders is higher in the case of severe psoriasis compared to its mild form [41]. An important issue to consider when evaluating the relationship between MDD and PV is the psychological impact of PV symptomatology on patients. Indeed, PV is characterized by skin lesions that affect body images and may lead to a decrease in patient’s self-esteem, self-confidence, and well-being, thereby contributing to the development of anxiety and depression [170]. In addition to these psychological factors, biological mechanisms, particularly systemic inflammation, may play a role in the development of MDD in PV patients. Indeed, current scientific evidence shows that pro-inflammatory cytokines are involved in its pathogenesis/progression and that inflammation involves the organism at a deeper level than skin. In moderate-to-severe psoriasis, for example, increased levels of pro-inflammatory markers and cytokines, such as TNF-α, IL-6, IL-17, and others, have been detected not only in skin plaques, but also in the blood and in other biological fluids, including saliva [41,170]. An important driver of pathogenesis in many inflammatory skin disorders, including psoriasis, is the activation of cutaneous T cells and dendritic cells and the upregulation of pro-inflammatory cytokines. Following migration to the skin and the release of pro-inflammatory cytokines, immune cells promote biological responses in keratinocytes, which themselves enhance further inflammation. Although inflammation may be local, arising in peripheral organs, pro-inflammatory cytokines such as TNF-α and IL-6, but also dendritic cells, can cross the blood-brain barrier and trigger a cascade of events at the central level, affecting for example the synaptic function via the actions of the microglia, neurotransmitter metabolism, and neurogenesis [171]. 

#### 3.2.4. Multiple Sclerosis

MS is the most frequently occurring neuroinflammatory disease, accompanied by demyelination, axonal and neuronal damage, with a wide range of clinical manifestations but an unknown etiology [172]. MS is characterized by elevated levels of pro-inflammatory cytokines, including IL-6 and IL-12, and decreased levels of anti-inflammatory agents, such as IL-10 [46]. Moreover, increased levels of chemokines, which stimulate leukocyte recruitment (i.e., CCL4, CCL22, and CXCL10) have been found, suggesting a subsequent T cell activation [47]. Interestingly, as reported by a recent systematic review and meta-analysis, patients with MS show a higher prevalence of developing MDD and experience typical depressive symptoms such as pain, fatigue, and cognitive impairment [173]. Moreover, altered cytokine profiles observed in MS patients are similar to those found in depressed patients [174]. For example, mRNA levels of *IFN*-γ and *TNF*-α are increased in MS patients and both cytokines are significantly correlated with depressive symptoms [48]. Similarly, a recent meta-analysis has reported significant associations between higher concentrations of TNF-α, IFN-γ, IL-6, and IL-10 and the severity of depressive symptoms in individuals with MS [175]. Several studies have also suggested the involvement of neuroinflammation, even subclinical, in MS-related mood symptoms. Higher rates and scores of MDD and anxiety have been observed among MS patients which decreased after brain inflammation resolution. In addition, MDD and anxiety have been suggested as hallmarks of active central inflammation in MS and suggest a cytokine-mediated pathogenic mechanism for comorbid affective disorders [176]. By evaluating, through neuroimaging studies, the cortical morphometric covariance networks of MS patients with and without depressive symptoms in comparison to a group of depressed patients without MS and control subjects, several similarities in the network reorganization between MS and MDD have been observed confirming both the involvement of neuroinflammation in patients with MDD and the increased vulnerability of MS patients to develop depressive symptoms [174].

All these findings corroborate the idea that inflammation should be considered as an important biological process that might increase the risk of mood disorders in MS patients and vice versa. 

#### 3.2.5. Inflammation as Both a Limit and a Target for AD Treatment in the Presence of Autoimmune Comorbidities 

In this context, where inflammation emerges as a shared mechanism across MDD and several autoimmune conditions, the immune system also plays a key role in influencing the treatment response. Indeed, in patients with MDD and comorbid autoimmune diseases, the response to ADs tends to be reduced due to the high inflammatory state characteristic of autoimmune conditions [41]. Considering this, the use of anti-inflammatory agents has gained increasing attention as potential strategy to alleviate depressive symptoms in individuals with a co-occurrent autoimmune disease. Among these, the administration of tocilizumab, a monoclonal antibody inhibiting IL-6, has been associated with a significant reduction of depressive symptoms in patients with RA [177]. Similarly, ustekinumab, a monoclonal antibody targeting IL-12 and IL-23, has been linked to mood improvement in patients with moderate-to-severe PV after 12 weeks of treatment [178].

### 3.3. Alterations in the Kynurenine Pathway and Cytokine Patterns Characterize Both Chronic Pain Conditions and MDD

Chronic pain is a highly prevalent condition in Europe, affecting approximately one-third of the population [179], and is a frequent comorbidity in MDD. Indeed, about 40-60% of individuals with MDD develop chronic pain [180], which is disabling in approximately 40% of cases [181]. Chronic pain syndrome is defined as pain that persists or recurs for at least three months in the absence of an identifiable medical condition. This definition encompasses several categories of pain [182], with low back pain (LBP), fibromyalgia, and migraine being among the most common comorbid ones in MDD patients [51]. 

Although the molecular mechanisms underlying chronic pain are not yet fully understood, inflammation has emerged as a key pathophysiological contributor. Indeed, in the presence of an inflammatory state, the release of pro-inflammatory mediators such as CXCL10, TNF-α, IL-1β, which bind to their receptors on nociceptive neurons, leads to peripheral sensitization and neuronal excitation, thereby contributing to the development of pain [183]. Interestingly, chronic pain conditions are more frequent in females than in males, with some pain-related diseases, such as endometriosis, affecting only females [184]. These differences in the prevalence of pain disorders are multifactorial and can be attributed to various factors, including anatomical differences and hormonal influences. Indeed, testosterone deficiency has been linked to a higher risk of inflammation of the nociceptive nervous system and, consequently, to chronic pain [184]. Moreover, inflammatory pathways contribute to chronic pain in a sex-specific way: in males, pain is primarily driven by microglia activation, while in females it is mediated by T cell activation [185].

#### 3.3.1. Low Back Pain

LBP is a leading cause of disability worldwide, and individuals with MDD are more likely to report LBP compared to the general population [186,187]. A major driver of LBP is intervertebral disc degeneration, caused by the aberrant release of pro-inflammatory molecules from nucleus pulposus and annulus fibrosus cells, as well as from infiltrating macrophages, T cells, and neutrophils [188]. Notably, pro-inflammatory molecules commonly upregulated in MDD, such as TNF-α, IL-1β, IL-6, CRP, are also elevated in LBP, supporting inflammation as a shared underlying mechanism [49,50].

#### 3.3.2. Fibromyalgia

Fibromyalgia, a severe chronic disease that involves widespread musculoskeletal pain, hyperalgesia, and allodynia, is characterized by alterations of inflammatory pathways. Indeed, dysregulation of the HPA axis in fibromyalgia leads to immune activation and increased release of pro-inflammatory cytokines, including TNF-α, IL-8, and IL-6, also implicated in MDD. Elevated levels of hs-CRP further support the presence of a low-grade inflammation, which may explain the increased risk of fibromyalgia among MDD patients and the high prevalence of MDD in individuals with fibromyalgia [51,52].

#### 3.3.3. Migraine

Migraine is another chronic pain condition frequently comorbid with MDD. Individuals with MDD have a three-fold increased risk of developing migraine, and MDD is an independent risk factor for both the onset and chronicity of migraine [189]. A central mechanism in migraine pathophysiology is the excessive release of calcitonin gene-related peptide (CGRP), a neuropeptide expressed in C-fibers of the trigeminal ganglion that modulate pain perception [190]. Pro-inflammatory mediators may trigger the CGRP release [191] and elevated CGRP levels have been observed in females with MDD, potentially explaining the higher prevalence of migraine in MDD patients compared to the general population. Similar to other chronic pain disorders, migraine is associated with systemic inflammation, characterized by increased levels of several cytokines such as TNF-α, IL-1β, IL-6, and IL-8 [53]. 

#### 3.3.4. Chronic Fatigue Syndrome and Endometriosis

Other two pain-related conditions that frequently co-occur with MDD are chronic fatigue/myalgic encephalomyelitis (CFS/ME) and endometriosis. Although not currently included in the International Classification of Diseases 11^th^ Revision as chronic diseases, these two disorders are often characterized by a persistent and unexpected pain. CFS/ME is characterized by prolonged fatigue in the absence of a clear medical cause and affects approximately 0.8-3.3% of the global population [192]. Adults with CFS/ME have a two-fold and half increased risk of developing MDD later in life, possibly due to a chronic inflammatory state that characterizes the disorder [193]. Indeed, several inflammatory markers, such as hs-CRP, CXCL-10, CCL24, IL-7, IL-8, IFN-γ, and TNF-α are elevated in the peripheral blood of CFS/ME patients and are positively correlated with symptom severity [54,55,56,57]. As in MDD, microglial activation is observed in CFS/ME and it is correlated with symptom severity [194].

Endometriosis is characterized by the diffusion of endometrial tissue outside the uterus and is diagnosed in approximately 10% of females of childbearing age [195]. Females with endometriosis have a one-fold and half increased risk of developing MDD compared to healthy controls and the severity of depressive symptoms has shown to correlate with the severity of endometriosis-related symptoms [196,197]. Endometriosis is increasingly being considered as a chronic pain disorder, as pain often persists even after the removal of underlying causes [198]. Endometriosis is characterized by a dysregulation in immune and inflammatory systems, which may contribute to its comorbidity with MDD. Indeed, an abnormal presence of activated macrophages, natural killer cells, T cells, and dendritic cells in endometrium promotes the establishment and progression of this debilitating disorder through the release of pro-inflammatory cytokines such as IL-1, IL-8, IL-33, and TNF-α, resulting in a pro-inflammatory microenvironment [58]. Moreover, IL-2 promotes migration of endometrial stromal cells outside the uterus, while IL-8 supports the survival, growth, invasion, differentiation, and immune evasion in ectopic sites [58].

#### 3.3.5. Involvement of Kynurenine Pathway in Chronic Pain Conditions

A further inflammatory mechanism linking MDD with several of the chronic pain conditions is the kynurenine pathway [19]. In migraine, both quinolinic and kynurenine acids can activate microglia and astrocytes, leading to the release of pro-inflammatory cytokines and chemokines in the central nervous system, thereby reinforcing inflammatory status [199]. Similarly, in patients with CFS/ME and fibromyalgia, the ratios between kynurenic and quinolinic acid, as well as between kynurenic acid and hydroxykynurenine, are lower than in controls, indicating a shift towards a neurotoxic and inflammatory state [52]. Moreover, in MDD patients, increased levels of quinolinic acid and reduced levels of tryptophan have been associated with greater pain severity [200].

#### 3.3.6. Chronic Pain Impacts on AD Successful Rate in MDD Patients 

The presence of a chronic pain condition in MDD complicates and reduces AD treatment response. For example, only 17% of MDD patients with co-morbid migraine achieve remission of depressive symptoms compared to 25% of those without migraine [201]. To make the picture even more complex, a reduced responsiveness in the presence of chronic pain comorbidities may also be influenced by polypharmacy, which is common among patients with both MDD and chronic pain. For instance, MDD patients with co-morbid LBP who are treated with both ADs and opioids show a lower response rate to ADs compared to those not taking opioids [202].

## 4. Conclusions

The role of inflammation in the pathophysiology of MDD and its impact on pharmacological treatment outcomes is well established. Indeed, numerous studies have documented altered inflammatory profiles in MDD patients, with elevated levels of pro-inflammatory markers, such as IL-1β, IL-6, TNF-α, and CRP. As evidenced in our narrative review, these same inflammatory mediators are also consistently upregulated in common MDD comorbidities, including cardiometabolic diseases, autoimmune disorders, and chronic pain-related conditions. This suggests that chronic low-grade inflammation may represent a shared biological mechanism underlying both MDD and its comorbidities. However, several comorbid conditions exhibit overlapping inflammatory profiles and upregulated cytokines, which makes it difficult, nowadays, to stratify patients based on their specific comorbidities (e.g., by identifying a cytokine uniquely associated with a given comorbidity). Therefore, the molecular mechanisms through which inflammation increases the risk of comorbidities in MDD remain poorly understood and the cause-and-effect mechanisms linking inflammation, MDD, and the co-occurrent disorders are still unclear.

The rising co-occurrence between MDD and all the diseases shows a growing global health challenge, with far-reaching clinical, social, and economic implications. Rather than being treated as distinct pathologies, MDD and its comorbidities may be seen as interconnected disorders, underpinned by dysregulated inflammatory pathways. Evidence that these conditions share chronic low-grade inflammation as a common pathophysiological pathway marks a critical shift in our understanding of their interconnectedness, highlighting new opportunities for early diagnosis, risk stratification, and targeted intervention. Despite these advances, important questions remain unanswered. Future research should focus on longitudinal and mechanistic investigations aimed at clarifying the temporal and causal mechanisms through which inflammation contributes to MDD and comorbid conditions. Additionally, interdisciplinary approaches integrating psychiatry, endocrinology, cardiology, immunology, and neurology will be essential to develop and test novel anti-inflammatory or immunomodulatory treatments. The reduced effectiveness of current treatments in the presence of inflammation and comorbidity underscores the urgent need for new clinical trials to longitudinally evaluate whether cytokine inhibitors, metabolic therapies, or anti-inflammatory lifestyle interventions (e.g., exercise, diet) can simultaneously alleviate depressive symptoms and improve outcomes in the co-occurring diseases. Moreover, taking sex differences into account is essential for the development of more effective and targeted treatments. In this context, inflammation not only emerges as a promising therapeutic target but also as a key factor for patient stratification and treatment personalization, to pave the way for more effective and tailored treatment for individuals suffering from MDD’s complicated comorbidities. 

## Figures and Tables

**Figure 1 ijms-26-09382-f001:**
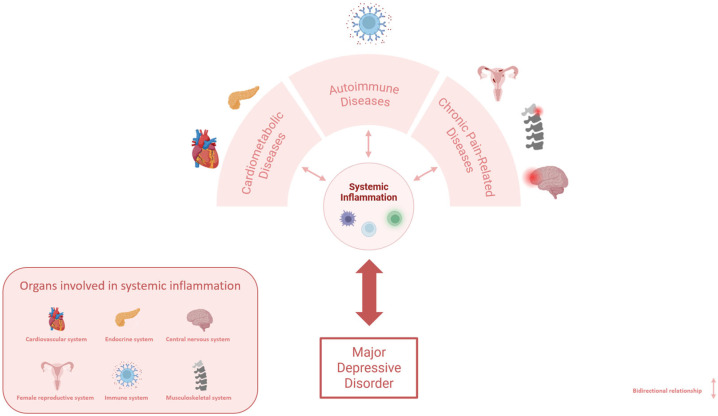
Bidirectional relationship between systemic inflammation, MDD, and comorbid conditions. Cardiometabolic, autoimmune, and chronic pain-related disorders can promote systemic inflammation, thereby increasing the risk of developing MDD. Conversely, MDD itself may trigger systemic inflammation, contributing to the onset of comorbid conditions.

## Data Availability

Not applicable.

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
