# Peer review of "Systemic Inflammation at the Crossroad of Major Depressive Disorder and Comorbidities: A Narrative Review"

_ijms, 2025, doi:10.3390/ijms26199382_

Round 1
Reviewer 1 Report
Comments and Suggestions for Authors
-
Inflammation and depression are bidirectionally linked; thus, inflammation could be a cause, consequence, or concomitant phenomenon in major depressive disorder (MDD). Since not all MDD patients exhibit elevated inflammation, could this heterogeneity relate to subtypes (e.g., 'inflammatory depression')? Caution is warranted when discussing the relationship between inflammation and depression and its comorbidities.
-
This review only explores the association between inflammatory markers and comorbidities without delving into deeper mechanisms, such as the gut-brain axis or epigenetics.
-
The structure needs refinement. It is unclear why the focus is limited to cardiometabolic diseases, autoimmune diseases, and chronic pain conditions—while sleep disorders, for instance, are also a critical comorbidity in MDD. I suggest first introducing the general landscape of MDD comorbidities, highlighting these three as among the most prevalent, before elaborating on inflammation’s role in MDD and these comorbid conditions.
-
I find the placement of Section 1.2 (Antidepressant treatments: how inflammation may modulate and predict their efficacy) structurally inappropriate. It would be better positioned as a third major section, shaping the review into a logical flow: (1) current understanding, (2) mechanistic links, and (3) interventions/therapeutic targets.
Author Response
1. Inflammation and depression are bidirectionally linked; thus, inflammation could be a cause, consequence, or concomitant phenomenon in major depressive disorder (MDD). Since not all MDD patients exhibit elevated inflammation, could this heterogeneity relate to subtypes (e.g., 'inflammatory depression')? Caution is warranted when discussing the relationship between inflammation and depression and its comorbidities.
We thank the reviewer for this interesting comment. We agree with the reviewer: as observed in section 1. Introduction: Role of inflammation in the pathophysiology of Major Depressive Disorder, approximately 60% of MDD patients exhibit mildly elevated inflammatory levels, in terms of increased CRP levels, and it is well established that inflammation is just one of the several mechanisms involved in the onset and treatment of this debilitating disorder. Indeed, not all patients show these changes, reflecting the heterogeneity of MDD. Moreover, the cause-and-effect relationship among inflammation, MDD, and its comorbidities remains poorly understood. Our review focuses on inflammatory biomarkers that could represent a link between MDD and these comorbidities. Thus, to better highlight this point, we have added the following sentence to section 4. Conclusion: “Therefore, the molecular mechanisms through which inflammation increases the risk of comorbidities in MDD remain poorly understood and the cause-and-effect mechanisms linking inflammation, MDD, and the co-occurrent disorders are still unclear.” (lines 661-664).
2. This review only explores the association between inflammatory markers and comorbidities without delving into deeper mechanisms, such as the gut-brain axis or epigenetics.
We thank the reviewer for this comment. The gut-brain axis and epigenetics are certainly important mechanisms that may link MDD with its comorbidities. However, since the main aim of our review is to provide an overview of the role of inflammation in the co-occurrence of MDD and the described pathologies, we did not explore these mechanisms in depth, but we agree that future reviews specifically addressing these aspects would be of great interest and complementary to our work.
3. The structure needs refinement. It is unclear why the focus is limited to cardiometabolic diseases, autoimmune diseases, and chronic pain conditions—while sleep disorders, for instance, are also a critical comorbidity in MDD. I suggest first introducing the general landscape of MDD comorbidities, highlighting these three as among the most prevalent, before elaborating on inflammation’s role in MDD and these comorbid conditions.
We thank the reviewer for this comment. Although we know that a lot of critical comorbidities for MDD exist, in the current review we specifically focused on cardiovascular diseases, diabetes, autoimmune and chronic-pain related conditions because these represent the most prevalent physical comorbidities in MDD and are strongly linked to inflammation, as discussed in our review. As suggested, we have highlighted this point in section 3. Inflammation as a triggering mechanism of comorbidities in MDD, by adding the following sentences:
- MDD is associated with an increased incidence of several comorbid conditions, including both physical (e.g. cardiovascular, metabolic, respiratory, oncological, neurological, gastroenterological, autoimmune, musculoskeletal, and chronic pain-related disorders) and psychiatric disorders (e.g. anxiety, eating and sleeping disorders, dysthymia, substance abuse, schizophrenia, and antisocial personality) [Cuomo et al., 2025; Fang et al., 2019; Thaipisuttikul et al., 2014; Arnaud et al., 2022]” (lines 229-233)
- “This narrative review explores the most frequent conditions co-occurring with MDD, namely cardiometabolic, autoimmune, and chronic pain-related disorders,” (lines 244-246)
4. I find the placement of Section 1.2 (Antidepressant treatments: how inflammation may modulate and predict their efficacy) structurally inappropriate. It would be better positioned as a third major section, shaping the review into a logical flow: (1) current understanding, (2) mechanistic links, and (3) interventions/therapeutic targets.
We thank the reviewer for this comment. We agree that section 1.2. Antidepressant treatments: how inflammation may modulate and predict their efficacy should be a major section. However, we believe it should be positioned after section 1. Introduction: Role of inflammation in the pathophysiology of Major Depressive Disorder, rather than after section 2. Inflammation as a triggering mechanism of comorbidities in MDD, since it focuses on targeting inflammatory mechanisms to treat MDD and on the effects of inflammation in modulating the antidepressant response, rather than on the role of comorbidities in MDD treatment. Therefore, Section 1.2. Antidepressant treatments: how the inflammation may modulate and predict their efficacy has now been repositioned as section 2. Antidepressant treatments: how the inflammation may modulate and predict their efficacy, and all other subsequent sections have been renumbered consequently.
References
Arnaud, A.M.; Brister, T.S.; Duckworth, K.; Foxworth, P.; Fulwider, T.; Suthoff, E.D.; Werneburg, B.; Aleksanderek, I.; Reinhart, M.L. Impact of Major Depressive Disorder on Comorbidities: A Systematic Literature Review. Journal of Clinical Psychiatry 2022, 83, E1–E12, doi:10.4088/JCP.21r14328.
Cuomo, A.; Koukouna, D.; Pardossi, S.; Pinzi, M.; Rescalli, M.B.; Pierini, C.; Fagiolini, A. Depression and Physical Comorbidities: An Integrated Review of Challenges and Treatment Approaches. Riv Psichiatr 2025, 60, 150–164, doi:10.1708/4548.45487.
Fang, H.; Tu, S.; Sheng, J.; Shao, A. Depression in Sleep Disturbance: A Review on a Bidirectional Relationship, Mechanisms and Treatment. J Cell Mol Med 2019, 23, 2324–2332, doi:10.1111/JCMM.14170.
Thaipisuttikul, P.; Ittasakul, P.; Waleeprakhon, P.; Wisajun, P.; Jullagate, S. Psychiatric Comorbidities in Patients with Major Depressive Disorder. Neuropsychiatr Dis Treat 2014, 10, 2097–2103, doi:10.2147/NDT.S72026.
Reviewer 2 Report
Comments and Suggestions for Authors
Major comments:
- Table 1 – please distinguish between studies that examine inflammatory gene expression and those that examine serum/plasma levels of inflammatory proteins, e.g., Osimo et al. reports changes in serum CRP, Das et al. examined serum IL-1beta and TNF, a similar thing concerns Min et al. 2023, Masubire et al. 2023, and several others.
- In Table 1, in the study by Meyer-Arndt et al. 2023, neurons were treated with proinflammatory cytokines, and cytokine receptor gene expression was examined.
- Also, you cite many articles about inflammatory proteins in comorbid conditions, but you write nothing about them in the Introduction. The information about them appears in Section 2.2, when some information about the coexistence of MDD and those conditions should be mentioned in the beginning.
- Sentence „MDD is associated with an increased incidence of several comorbid conditions, including cardiovascular, metabolic, autoimmune, and chronic pain disorders [48], which may interfere with AD treatment response [30] (Figure 1).” and other sections about comorbidities – we live in times where the majority of the population at a certain age will suffer from several chronic diseases such as CVD or diabetes, are MDD patients are actually more likely to suffer from CVD, diabetes or autoimmune diseases than people without depressive disorders? The authors mention something about it in a few paragraphs, but I would advise adding a separate section with detailed data on this subject.
- Section „1.2 Antidepressant treatments” – I would recommend expanding the paragraph by adding information about anti-inflammatory drugs (NSAIDs) or cytokine inhibitors and even curcumin, presenting strong anti-inflammatory properties:
- Köhler O, Benros ME, Nordentoft M et al (2014). Effect of antiinflammatory treatment on depression, depressive symptoms, and adverse effects a systematic review and meta-analysis of randomized clinical trials. JAMA Psychiatry 71:1381–1391,
- Schmidt FM, Kirkby KC, Himmerich H (2014b). The TNF-alpha inhibitor etanercept as monotherapy in treatment-resistant depression—report of two cases. Psychiatr Danub 26:288–290,
- Raison CL, Rutherford RE, Woolwine BJ et al (2013). A randomized controlled trial of the tumor necrosis factor antagonist infliximab for treatment-resistant depression: the role of baseline inflammatory biomarkers. JAMA Psychiatry 70:31–41,
- Yu JJ, Pei LB, Zhang Y et al (2015). Chronic supplementation of curcumin enhances the efficacy of antidepressants in major depressive disorder: a randomized, double-blind, placebo-controlled pilot study. J Clin Psychopharmacol 35:406–410.
- Sentence „Based on screening studies, MDD may affect up to 55% of patients with psoriasis, while the incidence of mood disorders is higher in the case of severe psoriasis compared to its mild form [122].” - whether a patient with psoriasis actually has depressive symptoms is the result of inflammation accompanying AD or emotions related to the disease itself, self-perception (psoriasis is a very serious disease that affects the patient's appearance)? Authors should discuss it.
- The authors write only one sentence about the relationships between pro-inflammatory cytokines and the HPA axis. This is a very important topic related to depression, which requires further exploration. It's worth noting the specific relationships between cytokines and the nervous system, the origins of cytokines in the nervous system, and the important role microglia play in inflammatory processes. It's also worth mentioning the role of glucocorticoid resistance in depression (Szałach ŁP, Lisowska KA, Cubała WJ. The Influence of Antidepressants on the Immune System. Arch Immunol Ther Exp (Warsz). 2019 Jun;67(3):143-151, Lee CH, Giuliani F. The Role of Inflammation in Depression and Fatigue. Front Immunol. 2019 Jul 19;10:1696.).
Minor comments:
- Please explain the shortcuts in Table 1: FKBP5 and GR.
- Samin et al. 2024 in Table 1 is not cited in the References.
- Section „2. Inflammation as a triggering mechanism of comorbidities in MDD” – I advise dividing this paragraph into sections devoted to various health issues (CVD, diabetes, etc.), which will be more readable.
- The article is very chaotic and disorganized, with some topics jumbled together. I suggest logical organization of the paragraphs and adding subsections to make the article easier to read.
Author Response
Major comments:
1. Table 1 – please distinguish between studies that examine inflammatory gene expression and those that examine serum/plasma levels of inflammatory proteins, e.g., Osimo et al. reports changes in serum CRP, Das et al. examined serum IL-1beta and TNF, a similar thing concerns Min et al. 2023, Masubire et al. 2023, and several others.
We thank the reviewer for this comment. Accordingly, we have modified Table 1 by distinguishing studies that investigated inflammatory gene expression from those that evaluated protein levels. Specifically, we separated proteins from genes by subdividing the original “Inflammation-related molecules” column into two different columns named “Inflammation-related proteins” and “Inflammation-related genes”.
2. In Table 1, in the study by Meyer-Arndt et al. 2023, neurons were treated with proinflammatory cytokines, and cytokine receptor gene expression was examined.
We thank the reviewer for bringing this oversight to our attention. We have replaced the reference with a more appropriate one and corrected the sentence in the paper that now reads “MS is characterized by elevated levels of pro-inflammatory cytokines, including IL-6 and IL-12, and decreased levels of anti-inflammatory agents, such as IL-10 [Kallaur et al., 2013]” (lines 479-482).
3. Also, you cite many articles about inflammatory proteins in comorbid conditions, but you write nothing about them in the Introduction. The information about them appears in Section 2.2, when some information about the coexistence of MDD and those conditions should be mentioned in the beginning.
We thank the reviewer for this comment. The following sentence has been now added in the section 1. Introduction: Role of inflammation in the pathophysiology of Major Depressive Disorder to introduce the coexistence of MDD with its co-occurrent conditions (lines 58-61): “The inflammatory hypothesis is one of the most studied mechanisms underpinning MDD and it is strongly supported by evidence that confirms its involvement in both MDD onset and antidepressant (AD) treatment response, and it has also been implicated in several common comorbidities associated with MDD.”. Moreover, a sentence has been added to introduce the investigation of inflammation-related molecules in the comorbid conditions taken into account in our review in section 3. Inflammation as a triggering mechanism of comorbidities in MDD (lines 249-251): “In particular, we will discuss inflammation-related molecules involved in MDD comorbidities to elucidate the presence of both shared and condition-specific inflammation-related biomarkers.”.
4. Sentence „MDD is associated with an increased incidence of several comorbid conditions, including cardiovascular, metabolic, autoimmune, and chronic pain disorders [48], which may interfere with AD treatment response [30] (Figure 1).” and other sections about comorbidities – we live in times where the majority of the population at a certain age will suffer from several chronic diseases such as CVD or diabetes, are MDD patients are actually more likely to suffer from CVD, diabetes or autoimmune diseases than people without depressive disorders? The authors mention something about it in a few paragraphs, but I would advise adding a separate section with detailed data on this subject.
We thank the reviewer for this interesting comment. MDD patients are more likely to develop comorbidities later in life compared to individuals without MDD, and the presence of MDD in adulthood may predict the development of such comorbidities. While this topic slightly deviates from the focus of our review, we acknowledge its relevance and, therefore, we have included detailed information in section 3. Inflammation as a triggering mechanism of comorbidities in MDD. In particular, the following sentence has been added: “Interestingly, the presence of MDD in adulthood may predict the onset of these disorders later in life. For instance, during aging, MDD patients have an increased risk of developing neurological disorders such as Parkinson’s disease and Alzheimer’s disease [Schäbitz et al., 2003; Sáiz-Vázquez et al., 2021], as well as cardiovascular conditions [O’neil et al., 2016]. Additionally, late-life onset of MDD has been associated with a higher risk of several aforementioned co-occurrent conditions [Lyness et a., 2006; Grover et al., 2017; Diniz et al., 2013], which may contribute to elevated mortality in patients with more severe depressive symptoms [Diniz et al., 2014]” (lines 236-242).
5. Section „1.2 Antidepressant treatments” – I would recommend expanding the paragraph by adding information about anti-inflammatory drugs (NSAIDs) or cytokine inhibitors and even curcumin, presenting strong anti-inflammatory properties:
6. Köhler O, Benros ME, Nordentoft M et al (2014). Effect of antiinflammatory treatment on depression, depressive symptoms, and adverse effects a systematic review and meta-analysis of randomized clinical trials. JAMA Psychiatry 71:1381–1391,
7. Schmidt FM, Kirkby KC, Himmerich H (2014b). The TNF-alpha inhibitor etanercept as monotherapy in treatment-resistant depression—report of two cases. Psychiatr Danub 26:288–290,
8. Raison CL, Rutherford RE, Woolwine BJ et al (2013). A randomized controlled trial of the tumor necrosis factor antagonist infliximab for treatment-resistant depression: the role of baseline inflammatory biomarkers. JAMA Psychiatry 70:31–41,
9. Yu JJ, Pei LB, Zhang Y et al (2015). Chronic supplementation of curcumin enhances the efficacy of antidepressants in major depressive disorder: a randomized, double-blind, placebo-controlled pilot study. J Clin Psychopharmacol 35:406–410.
We thank the reviewer for this interesting suggestion. We have added the following sentences regarding the use of infliximab, adalimumab, etanercept, tocilizumab, and curcumin in section 2. Antidepressant treatments: how the inflammation may modulate and predict their efficacy: “Inhibition of inflammatory cytokines has also been evaluated as a strategy to improve depressive symptoms. For instance, the adjunctive use of infliximab, a monoclonal antibody targeting TNF-α, improved HAMD-17 scores in patients with hs-CRP > 5 mg/L, and decreased inflammation levels [Raison et al., 2013]. Similarly, adalimumab, a monoclonal antibody that binds to TNF-α, when combined with sertraline, induced a reduction of HAM-D score in MDD patients [Abbasian et al., 2022]. Also, etanercept and tocilizumab, already used in autoimmune diseases, are able to improve depressive symptoms in patient with psoriasis and rheumatoid arthritis, respectively [Jin et al., 2019; Tiosano et al., 2020; Kappelmann et al., 2018]. In addition to pharmacological anti-inflammatory agents, natural compounds with strong anti-inflammatory properties, such as curcumin, have also been investigated. Curcumin, a compound derived from the Curcuma longa plant, has interesting anti-inflammatory properties and can modulate neurotransmitter concentration, inflammatory pathways, neuroplasticity, HPA axis impairment, and other mechanisms involved in MDD [Ramaholimihaso et al., 2020]. Notably, MDD patients treated with curcumin showed a decrease in both HDRS-17 and MADRS score compared with the placebo group, along with a reduction of pro-inflammatory cytokines [Yu et al., 2019].” (lines 182-197).
10. Sentence „Based on screening studies, MDD may affect up to 55% of patients with psoriasis, while the incidence of mood disorders is higher in the case of severe psoriasis compared to its mild form [122].” - whether a patient with psoriasis actually has depressive symptoms is the result of inflammation accompanying AD or emotions related to the disease itself, self-perception (psoriasis is a very serious disease that affects the patient's appearance)? Authors should discuss it.
We thank the reviewer for this very insightful comment. We agree that psoriasis vulgaris (PV) may impact on patient’s mood due to its symptomatology. We have discussed this in section 3.2.3 Psoriasis Vulgaris to highlight this important issue: “An important issue to consider when evaluating the relationship between MDD and PV is the psychological impact of PV symptomatology on patients. Indeed, PV is characterized by skin lesions that affect body images and may lead to a decrease in patient’s self-esteem, self-confidence, and well-being, thereby contributing to the development of anxiety and depression [Wintermann et al., 2023]” (lines 451-455).
11. The authors write only one sentence about the relationships between pro-inflammatory cytokines and the HPA axis. This is a very important topic related to depression, which requires further exploration. It's worth noting the specific relationships between cytokines and the nervous system, the origins of cytokines in the nervous system, and the important role microglia play in inflammatory processes. It's also worth mentioning the role of glucocorticoid resistance in depression (Szałach ŁP, Lisowska KA, Cubała WJ. The Influence of Antidepressants on the Immune System. Arch Immunol Ther Exp (Warsz). 2019 Jun;67(3):143-151, Lee CH, Giuliani F. The Role of Inflammation in Depression and Fatigue. Front Immunol. 2019 Jul 19;10:1696.).
We thank the reviewer for this comment and agree with the need of including in our review a brief description of the role of the HPA axis in MDD and its interplay with microglial functioning. Thus, we have added the following paragraph in section 1. Introduction: Role of inflammation in the pathophysiology of Major Depressive Disorder: “
Microglial activation is also closely related to another mechanism implicated in MDD, namely the hypothalamic-pituitary-adrenal (HPA) axis [Cheiran et al., 2022]. Under physiological conditions, the HPA axis is activated in response to stress, which induces the release of corticotropinreleasing hormone (CRH) from the hypothalamus and, consequently, the secretion of adrenocorticotropic hormone (ACTH) from anterior pituitary gland. ACTH, in turn, stimulates the release of glucocorticoids (GCs), in particular cortisol, from the adrenal cortex. Cortisol acts through a negative feedback loop, inhibiting further secretion of CRH and ACTH by binding to GRs in the hypothalamus and pituitary gland [Mikulska et al., 2021]. Under chronic stress, this mechanism is disrupted and the continuous production of cortisol desensitizes GRs, leading to GC resistance and aberrant HPA axis activation [Mikulska et al., 2021]. In the presence of neuroinflammation, microglial activation can further stimulate HPA axis activity. Indeed, pro-inflammatory cytokines released by microglia, including IL1-β, IL-6, IFN-α and TNF-α, triggers CRH secretion from the hypothalamus, resulting in HPA axis hyperactivation [Cheiran et al., 2022]. In turn, the release of GCs may contribute to neuroinflammation through the binding of cortisol to mineralocorticoid receptors and GRs expressed on the microglia. In particular, excessive levels of GCs promote the pro-inflammatory activation of GR-rich microglial cells, thus sustaining the neuroinflammatory status [Cheiran et al., 2022].” (lines 97-112).
Minor comments:
1. Please explain the shortcuts in Table 1: FKBP5 and GR.
We thank the reviewer for this comment. We have added the full acronym of FKBP5 “FKBP prolyl isomerase 5” in line 158-159. The full name of GR was already provided in the manuscript when the acronym was first introduced (line 72).
2. Samin et al. 2024 in Table 1 is not cited in the References.
We thank the reviewer for pointing this out. The citation “Samin et al., 2024” has been removed from Table 1.
3. Section „2. Inflammation as a triggering mechanism of comorbidities in MDD” – I advise dividing this paragraph into sections devoted to various health issues (CVD, diabetes, etc.), which will be more readable.
We thank the reviewer for this comment. We have reorganized the section 2. Inflammation as a triggering mechanism of comorbidities in MDD (now section 3. Inflammation as a triggering mechanism of comorbidities in MDD) on comorbidities discussing each condition separately to make the paper more readable.
4. The article is very chaotic and disorganized, with some topics jumbled together. I suggest logical organization of the paragraphs and adding subsections to make the article easier to read.
We thank the reviewer for bringing this important point to our attention. Following the reviewers’ suggestion, we have refined Table1, reorganized the text by discussing each comorbidity in a dedicated paragraph, and added new subsections where appropriate. We believe these changes have improved the logical flow of the manuscript and made it more readable.
References
Abbasian, F.; Bagheri, S.; Moradi, K.; Keykhaei, M.; Etemadi, A.; Shalbafan, M.; Shariati, B.; Vaseghi, S.; Samsami, F.S.; Akhondzadeh, S. Evidence for Anti-Inflammatory Effects of Adalimumab in Treatment of Patients With Major Depressive Disorder: A Pilot, Randomized, Controlled Trial. Clin Neuropharmacol 2022, 45, 128–134, doi:10.1097/WNF.0000000000000518.
Cheiran Pereira, G.; Piton, E.; Moreira dos Santos, B.; Ramanzini, L.G.; Muniz Camargo, L.F.; Menezes da Silva, R.; Bochi, G.V. Microglia and HPA Axis in Depression: An Overview of Participation and Relationship. World J Biol Psychiatry 2022, 23, 165–182, doi:10.1080/15622975.2021.1939154.
Diniz, B.S.; Butters, M.A.; Albert, S.M.; Dew, M.A.; Reynolds, C.F. Late-Life Depression and Risk of Vascular Dementia and Alzheimer’s Disease: Systematic Review and Meta-Analysis of Community-Based Cohort Studies. Br J Psychiatry 2013, 202, 329–335, doi:10.1192/BJP.BP.112.118307.
Diniz, B.S.; Reynolds, C.F.; Butters, M.A.; Dew, M.A.; Firmo, J.O.A.; Lima-Costa, M.F.; Castro-Costa, E. The effect of gender, age, and symptom severity in late-life depression on the risk of all-cause mortality: the bambuí cohort study of aging. Depress Anxiety 2014, 31, 787–795, doi:10.1002/DA.22226.
Grover, S.; Dalla, E.; Mehra, A.; Chakrabarti, S.; Avasthi, A. Physical Comorbidity and Its Impact on Symptom Profile of Depression among Elderly Patients Attending Psychiatry Services of a Tertiary Care Hospital. Indian J Psychol Med 2017, 39, 450, doi:10.4103/0253-7176.211764.
Jin, W.; Zhang, S.; Duan, Y. Depression Symptoms Predict Worse Clinical Response to Etanercept Treatment in Psoriasis Patients. Dermatology 2019, 235, 55–64, doi:10.1159/000492784.
Kallaur, A.P.; Oliveira, S.R.; Simao, A.N.C.; De Almeida, E.R.D.; Morimoto, H.K.; Lopes, J.; De Carvalho Jennings Pereira, W.L.; Andrade, R.M.; Pelegrino, L.M.; Borelli, S.D.; et al. Cytokine Profile in Relapsing-Remitting Multiple Sclerosis Patients and the Association between Progression and Activity of the Disease. Mol Med Rep 2013, 7, 1010–1020, doi:10.3892/MMR.2013.1256/HTML.
Kappelmann, N.; Lewis, G.; Dantzer, R.; Jones, P.B.; Khandaker, G.M. Antidepressant Activity of Anti-Cytokine Treatment: A Systematic Review and Meta-Analysis of Clinical Trials of Chronic Inflammatory Conditions. Mol Psychiatry 2018, 23, 335–343, doi:10.1038/MP.2016.167.
Lyness, J.M.; Niculescu, A.; Tu, X.; Reynolds, C.F.; Caine, E.D. The Relationship of Medical Comorbidity and Depression in Older, Primary Care Patients. Psychosomatics 2006, 47, 435–439, doi:10.1176/APPI.PSY.47.5.435.
Mikulska, J.; Juszczyk, G.; Gawrońska-Grzywacz, M.; Herbet, M. HPA Axis in the Pathomechanism of Depression and Schizophrenia: New Therapeutic Strategies Based on Its Participation. Brain Sci 2021, 11, doi:10.3390/BRAINSCI11101298.
O’neil, A.; Fisher, A.J.; Kibbey, K.J.; Jacka, F.N.; Kotowicz, M.A.; Williams, L.J.; Stuart, A.L.; Berk, M.; Lewandowski, P.A.; Taylor, C.B.; et al. Depression Is a Risk Factor for Incident Coronary Heart Disease in Women: An 18-Year Longitudinal Study. J Affect Disord 2016, 196, 117–124, doi:10.1016/j.jad.2016.02.029.
Raison, C.L.; Rutherford, R.E.; Woolwine, B.J.; Shuo, C.; Schettler, P.; Drake, D.F.; Haroon, E.; Miller, A.H. A Randomized Controlled Trial of the Tumor Necrosis Factor Antagonist Infliximab for Treatment-Resistant Depression: The Role of Baseline Inflammatory Biomarkers. JAMA Psychiatry 2013, 70, 31–41, doi:10.1001/2013.JAMAPSYCHIATRY.4.
Ramaholimihaso, T.; Bouazzaoui, F.; Kaladjian, A. Curcumin in Depression: Potential Mechanisms of Action and Current Evidence-A Narrative Review. Front Psychiatry 2020, 11, doi:10.3389/FPSYT.2020.572533.
Sáiz-Vázquez, O.; Gracia-García, P.; Ubillos-Landa, S.; Puente-Martínez, A.; Casado-Yusta, S.; Olaya, B.; Santabárbara, J. Depression as a Risk Factor for Alzheimer’s Disease: A Systematic Review of Longitudinal Meta-Analyses. J Clin Med 2021, 10, 1809, doi:10.3390/JCM10091809.
Schäbitz, W.-R.; Glatz, K.; Schuhan, C.; Sommer, C.; Berger, C.; Schwaninger, M.; Hartmann, M.; Goebel, H.H.; Meinck, H.-M. Higher Incidence of Depression Preceding the Onset of Parkinson’s Disease: A Register Study. Movement Disorders 2003, 18, 414–418, doi:10.1002/MDS.10387.
Tiosano, S.; Yavne, Y.; Watad, A.; Langevitz, P.; Lidar, M.; Feld, J.; Tishler, M.; Aamar, S.; Elkayam, O.; Balbir-Gurman, A.; et al. The Impact of Tocilizumab on Anxiety and Depression in Patients with Rheumatoid Arthritis. Eur J Clin Invest 2020, 50, doi:10.1111/ECI.13268.
Wintermann, G.B.; Bierling, A.L.; Peters, E.M.J.; Abraham, S.; Beissert, S.; Weidner, K. Psychosocial Stress Affects the Change of Mental Distress under Dermatological Treatment-A Prospective Cohort Study in Patients with Psoriasis. Stress Health 2023, 40, e3263–e3263, doi:10.1002/SMI.3263.
Yu, J.J.; Pei, L.B.; Zhang, Y.; Wen, Z.Y.; Yang, J.L. Chronic Supplementation of Curcumin Enhances the Efficacy of Antidepressants in Major Depressive Disorder: A Randomized, Double-Blind, Placebo-Controlled Pilot Study. J Clin Psychopharmacol 2015, 35, 406–410, doi:10.1097/JCP.0000000000000352.
Round 2
Reviewer 2 Report
Comments and Suggestions for Authors
None.
Author Response
We thank the Reviewer for suggesting an improvement of the English. All the authors have carefully revised the manuscript to improve the language.